# Characteristics of Organic Phosphorus Pool in Soil of Typical Agriculture Systems in South China

**Tong Li [1,2], Jianwu Yao [2], Ruikun Zeng [2], Yong Chen [2], Lijiang Hu [2], Mengyao Zou [1] and Jianfeng Ning [2,*]**

1 College of Resources and Environment, Zhongkai University of Agriculture and Engineering, Guangzhou 510550, China
2 Key Laboratory of Plant Nutrition and Fertilizer in South Region, Ministry of Agriculture, Guangdong Key Laboratory of Nutrient Cycling and Farmland Conservation, Institute of Agricultural Resources and Environment, Guangdong Academy of Agricultural Sciences, Guangzhou 510640, China
* Correspondence: jianfengning@gdaas.cn

**Abstract:** Organic phosphorus (P) is an important potential source of plant P nutrition in agro-ecosystems. It was hypothesized that the soil organic P pools were distinguished one from another by different land-utilization patterns. A total of 38 sites were sampled, to clarify the organic P pool and its distribution in paddy fields, vegetable fields, and orchards. Soil organic P fractions, including labile organic P (LOP), moderately labile organic P (MLOP), moderately resistant organic P (MROP) and highly resistant organic P (HROP) were examined. Results showed that the soil total P (TP) and available P (AP) concentration have enhanced by 138% and 1559%, respectively, over the last four decades. The soil total organic P (TOP) accounted for 21.4% of the TP pool. Soil MLOP dominated the organic P reservoir, irrespective of land-planting pattern. Soil organic P fractions ranked as MLOP > MROP > HROP > LOP. The highest accumulations of TP, AP and TOP were in the vegetable fields, followed by orchards and paddy fields. The vegetable fields had higher LOP and MLOP levels than those of the paddy fields and orchards, whereas the paddy fields exhibited higher concentrations of MROP, and HROP. Soil pH, organic matter and available nitrogen all contributed to the buildup of the organic P pool. It was suggested that soil organic P should be considered preferentially in the management of the plant P nutrient in regional planting systems.

**Keywords:** organic phosphorus; latosolic red soil; orchard; paddy field; vegetable field; phosphorus fractions

## 1. Introduction

Phosphorus (P) is an essential macronutrient, and plays a vital role in plant growth and crop production in agro-ecosystems. The application of P fertilizer is widely adopted to meet increasing food demands under the rapid development of populations and economies. Generally, P availability in soil is low, because most added P is fixed by iron and aluminum oxides, and clay minerals, resulting in a large surplus of P in soil over the long term [1]. In China, the consumption of phosphate fertilizers increased from 2.96 Tg in 1981 to 7.29 Tg in 2018 [2], accompanied with an extensive accumulation of P, termed as legacy P, in agricultural fields, i.e., an average of 242 kg P hm$^{-2}$ of soil legacy P from 1980 to 2007 [3]. However, the phosphorus used in fertilizer is extracted from rock phosphate, which is a nonrenewable resource that could be depleted in 50–100 years [4]. Therefore, rational exploitation of soil legacy P is imperative to reduce the P chemical fertilizer input on crops and alleviate the P scarcity crisis [5]. It is estimated that the legacy P could theoretically meet crop demands for approximately 9–22 years globally, if it were available [6].

Organic P compounds in soil vary enormously, but often constitute a large part of the total legacy P in surface soil, ranging from 20 to 80% in most soils [7]. Organic P compounds are an important component in the maintenance of P supply to crops from soil, contributing to plant P nutrition through the processes of mineralization [8]. Measuring Olsen P in

soil extractions is used to estimate soil P availability for plant uptake [9]. However, it has often failed to provide a comprehensive index of soil P fertility [10]. Thus, to ensure a deep understanding of soil P availability and the characteristics of the organic P pool, a more complete picture of the P is needed. Various approaches have been developed to reveal the forms, amount and dynamics of the P cycle. An example of this is the fractionation developed by Bowman and Cole (1978) [11], which is the sequential extraction of P from soil, which offers a measure of organic P pools with decreasing availability to plants. Sequential organic P fractionation distinguishes labile organic P (LOP), moderately labile organic P (MLOP), moderately resistant organic P (MROP) and highly resistant organic P (HROP) [11]. The chemical fractionation method evaluates the location and bonding type of P within the soil matrix, providing valuable information for the P speciation and dynamics in natural and managed systems [12]. Until now, wet chemical extraction schemes were still considered an important way to estimate the P speciation in soils although the solution $^{31}P$ nuclear magnetic resonance ($^{31}P$ NMR) spectroscopy has also been introduced to study the transformations of organic P in ecosystems.

Historically, rice cultivation dominated agricultural production in Guangdong Province of South China. In recent decades, however, with the regulations of the agriculture industrial structure, the planting of economic crops (vegetables and fruits) increased considerably, especially in suburban areas of cities. Guangzhou is the capital city of Guangdong Province, South China. In 2020, the sown areas of rice, vegetables and fruits in the city were recorded as 28,157, 150,974 and 70,418 hm$^2$, respectively [13]. It was estimated that rice and vegetables accounted for 10.3 and 70.2% of the annual total sown area of farm crops, respectively. Generally, paddy and vegetable fields, and orchards, are the main agriculture in the city, and also the typical land utilization distributed in South China. The paddy field is cultivated with two rice-growing seasons annually, which are 'early rice'(March to July), and 'late rice'(August to November). However, the vegetable field is often tilled to produce three to ten crops throughout the year. In contrast, the local orchards are usually planted with perennial tropical trees including litchi, longan, wampee trees, and so on. The three planting systems (paddy field, vegetable field and orchard) differ from each other in cultivation management. Agronomic practices influence the soil P balance, as well as the P availability and stability [6], implying that the soil organic P pool in paddy fields, vegetable fields, and orchards, may have their own characteristics. Therefore, the measurement and analysis of the organic P pool and its fractions are important, to improve the estimation of P plant availability, as well as P fertilizer recommendations across a range of agricultural crops and tillage management methods [14]. Until now, little information was available on the characteristics of the soil organic P pool between different land utilization pattens, i.e., paddy fields, vegetable fields and orchards.

The purpose of this study were to (i) reveal the current status of the soil total P and the available P in paddy fields, vegetable fields and orchards, and (ii) clarify the soil organic P pool, and its distribution in different agricultural environments. The results may provide theoretical references for improving the utilization and management of soil P in typical planting systems. This is vital for saving P rock resources and ensuring the sustainable development of the agricultural industry.

## 2. Materials and Methods

### 2.1. Study Area and Soil Collection

The study area (112°58' E–113°24' E, 23°19' N–23°33' N) is situated in the suburb of Guangzhou city, Guangdong province, South China. The site has an area of 970.04 km$^2$ and has a southern subtropical monsoon climate, with mean annual temperature of 22.9 °C and mean annual precipitation of 2376.7 mm during the last 22 years. Approximately 80% of the annually precipitation events occur in the periods from April to the September. The research area has a long history of agriculture cropping and fruit plantations, with intensive fertilization managements. The typical cultivation pattern in the land of the research area is characterized as rice, vegetable and fruit planting throughout the year. For the paddy

fields, rice is planted in two seasons yearly, i.e., early rice from March to July, late rice from Autumn to November. In the vegetable fields, various vegetables including leafy vegetables, gourd vegetables, rhizome vegetables, and beans, as well as solanaceous fruit vegetables, were planted with random rotation annually. In contrast, the local orchards are usually planted with perennial tropical trees, such as litchi, longan, wampee and so on. Some farmlands in the research area are usually cultivated with a rotation model, i.e., rice and vegetables, potato or other crops in rotation annually. In order to explore the influence of typical agriculture practices on the soil organic pool, the farmland with the following features was selected for this study: annual tillage with single-planting mode for more than five years, i.e., a paddy field with double cropping rice (early rice and late rice), vegetables field with year-round vegetable cultivation, and an orchard planted with perennial fruit trees. According to the distribution of crops and fruits in the study area, 38 soil sites including 12 paddy field sites, 15 vegetable field sites and 11 orchard sites were sampled during June to July 2021 (Figure 1). In each site, five soil cores (diameter = 3.0 cm) were taken to mix into one soil sample from the surface soil layer (0–20 cm). A total of 38 surface soil samples were collected from the 38 sites in the study area. The fresh soil samples were air dried after the removal of the crop or grass roots and gravels. The soil samples were ground to pass through 2 mm and 0.15 mm mesh, for soil analysis.

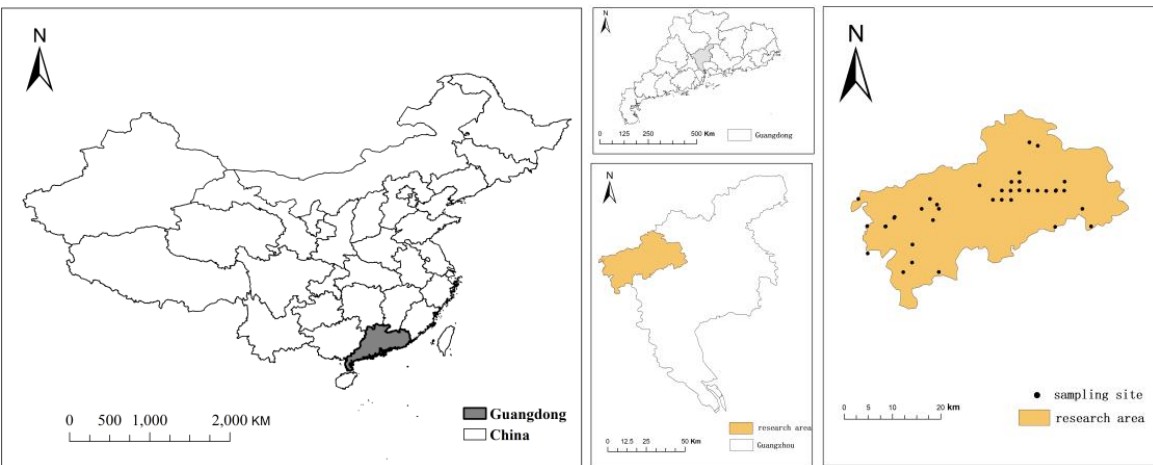

**Figure 1.** Location of the study area and sampling sites.

### 2.2. Laboratory Analysis

#### 2.2.1. General Chemical Properties

Basic chemical properties of the soil were determined according to the method of Lu (2000) [15]. Soil pH was measured using a pH meter (FE28-CN, Mettler-Toledo, Shanghai, China) at a soil-water ratio of 1:2.5. The organic carbon content was assayed with the potassium dichromate-sulfuric acid method, and then converted to soil organic matter (OM) by multiplying by 1.724. The available nitrogen (N) content in the soil sample was analyzed with the alkaline solution diffusion method. The available potassium in the soil was extracted with 1.0 M ammonium acetate, and the extraction was determined by using a flame atomic absorption spectrophotometer (ZA-3300, Hitachi, Tokyo, Japan). The soil total P content was digested with the HF-HClO$_4$ mixture, and analyzed with the molybdenum-blue colorimetry method. The available P (Olsen P) was detected with the method of Olsen (1954), as described by Lu (2000) [15]. The data of TP and AP were expressed as mg P kg$^{-1}$.

#### 2.2.2. Soil Organic P Fractions

According to the method developed by Bowman and Cole (1978) [11], the soil organic P was divided into four fractions, using a sequential extraction procedure: (1) labile organic P (LOP), extracted with 0.05 M NaHCO$_3$ (pH 8.5), with organic P determined by the difference between TP and inorganic P of the extractant; (2) MLOP, extracted with 1.0 M

$H_2SO_4$, with organic P determined in a similar way to that of LOP; (3) MROP, the fulvic acid P and (4) HROP, the humic acid P, both extracted with 0.5 M NaOH. The MROP and HROP were firstly determined together in the solution of 0.5 M NaOH extract, digested by perchloric acid (Solution A). An identical 0.5 M NaOH extract then received concentrated hydrochloric acid, to adjust the pH to between 1 and 1.8, and MROP was analyzed after perchloric acid digestion (Solution B). The HROP was calculated by deducting Solution B from Solution A. The P concentration in all extracts was assayed colorimetrically with the method of Murphy and Riley (1962) [16]. The total organic P concentrations were calculated from the sum of the LOP, MLOP, MROP and HROP. All the data of organic P fractions were expressed as mg P $kg^{-1}$.

### 2.3. Statistical Analysis

Data were expressed as the means of the three replicates. Correlation analysis was performed with IBM SPSS statistics versions 22.0. All figures were plotted using OriginPro 2021 (OriginLab Corporation., Northampton, MA, USA).

### 3. Results
#### 3.1. Soil Properties

Table 1 shows the soil properties. The soils were generally slightly acidic; the pH varied between the paddy fields, vegetable fields and orchards, at 5.32–7.90, 3.98–6.91, and 3.79–7.16, respectively. Average soil OM concentration (and ranges) was 16.12 (9.77–25.41) g·$kg^{-1}$, 18.89 (9.12–30.26) g·$kg^{-1}$, and 20.01 (9.07–29.78) g·$kg^{-1}$, in the paddy fields, vegetable fields and orchards, respectively. Average soil AN concentration (and ranges) was 76.29 (33.53–123.42) mg·$kg^{-1}$, 114.8 (61.16–152.53) mg·$kg^{-1}$, and 119.8 (54.53–249.79) mg·$kg^{-1}$, for the paddy fields, vegetable fields and orchards, respectively. Soil mean available potassium (AK) concentration was 87.43 (58.61–149.04) mg·$kg^{-1}$, 173.14 (50.17–291.75) mg·$kg^{-1}$, and 160.83 (63.32–312.17) mg·$kg^{-1}$, in the paddy fields, vegetable fields and orchards, respectively. The whole research area had an average pH of 5.73, and average concentration of OM, AN, and AK, were 18.34 g·$kg^{-1}$, 104.08, and 142.51 mg·$kg^{-1}$, respectively. Based on the soil nutrient classification standards (Table S1) [17], soil OM and AN were both in the medium level (grade 3 and 4 level), whereas AK concentrations were in the medium to rich levels (grade 2 to 4 level).

**Table 1.** The basic chemical properties in soil of different planting systems.

| Planting System | Parameters | pH | Organic Matter (OM) (g·$kg^{-1}$) | Available N (AN) (mg·$kg^{-1}$) | Available K (AK) (mg·$kg^{-1}$) |
|---|---|---|---|---|---|
| Paddy field | Mean | 6.47 | 16.12 | 76.29 | 87.43 |
| | Range | 5.32–7.90 | 9.77–25.41 | 33.53–123.42 | 58.61–149.04 |
| | C.V.(%) | 13.51 | 32.59 | 39.97 | 31.08 |
| Vegetable field | Mean | 5.32 | 18.89 | 114.8 | 173.14 |
| | Range | 3.98–6.91 | 9.12–30.26 | 61.16–152.53 | 50.17–291.75 |
| | C.V.(%) | 14.66 | 30.18 | 23.02 | 38.79 |
| Orchard | Mean | 5.48 | 20.01 | 119.8 | 160.83 |
| | Range | 3.79–7.16 | 9.07–29.78 | 54.53–249.79 | 63.32–312.17 |
| | C.V.(%) | 14.66 | 30.18 | 23.02 | 38.79 |

Note: C.V., coefficient variation.

#### 3.2. Soil Total P, Available P and Organic P

The mean soil TP concentration in the paddy fields, vegetable fields and orchards, was 627.85, 1255.26 and 731.54 mg·$kg^{-1}$, respectively (Figure 2). Overall, soil TP concentration in the surveyed area ranged from 230.37 to 2015.05 mg·$kg^{-1}$, with a mean value of 905.53 mg·$kg^{-1}$ (Figure 2), corresponding to a medium status, as defined by the soil nutrient classification standards [17]. Based on data surveyed in 1980 [18], the soil TP in the farmlands of Guangdong Province was 379.86 mg·$kg^{-1}$ (equivalent to 0.087% $P_2O_5$),

suggesting that soil TP concentration has increased by 138.4% over the past four decades. The soil TP concentration was ranked as follows: vegetable fields > orchards > paddy fields.

The average soil AP concentration (and ranges) was 51.03 (10.63–135.26), 153.11 (64.66–234.69) and 105.86 (50.25–261.34) mg·kg$^{-1}$, for the paddy fields, vegetable fields and orchards, respectively (Figure 2). In the research area, the average concentration of soil AP was 107.19 mg·kg$^{-1}$, indicating a rich status, based on the soil nutrient classification standards [17]. It was estimated that the soil AP concentration has increased by 1559% from 1980 to 2020 (i.e., soil AP concentration of 14.8 mg·kg$^{-1}$ ($P_2O_5$) in farmland of Guangdong province in 1980 [18]). The highest concentration of AP was in the vegetable fields, followed by the orchards and paddy fields, which is similar to that of TP (Figure 2).

Total organic P concentrations in the soil of the three different planting systems are displayed in Figure 2. In the paddy fields, the soil TOP ranged from 46.15 to 326.19 mg·kg$^{-1}$, with an average concentration of 169.71 mg·kg$^{-1}$. The vegetable fields showed a similar TOP concentration range of 46.96 to 385.34 mg·kg$^{-1}$, with an average concentration of 200.08 mg·kg$^{-1}$. The TOP in the soil of the orchards ranged from 73.04 to 245.84 mg·kg$^{-1}$, and the mean concentration of TOP was 137.30 mg·kg$^{-1}$. which was lower than that of the paddy fields and the vegetable fields. The TOP concentration in the research area was 172.32 mg·kg$^{-1}$, and ranked in the order of vegetable fields > paddy fields > orchards (Figure 2).

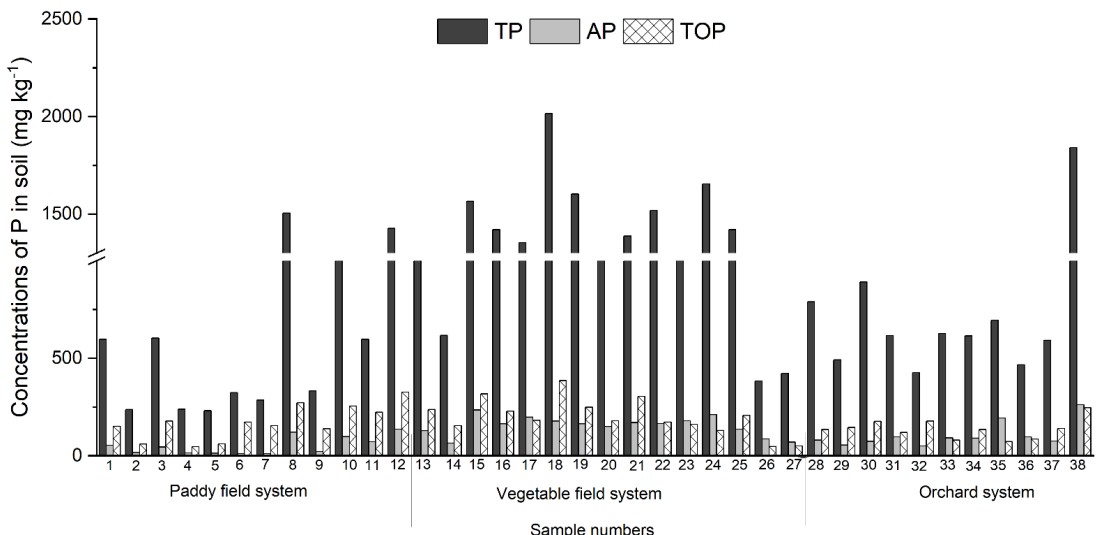

**Figure 2.** Concentrations of phosphorus in soil of different planting systems. Note: TP, total phosphorus; AP, available phosphorus; TOP, total organic phosphorus.

### 3.3. Soil Organic P Fractions

### 3.3.1. Labile Organic P (LOP)

The soil labile organic P in the three different planting systems are shown in Figure 3. The average concentration of LOP in the paddy fields, vegetable fields and orchards was 5.01,10.04 and 7.43 mg·kg$^{-1}$, respectively. There were wide variations in LOP values, as follows: 0.97–12.48, 2.15–18.1, 2.23–16.2 mg·kg$^{-1}$ with coefficients of variation of 75.17, 50.16 and 58.68%, in the paddy fields, vegetable fields, and orchards, respectively. The average LOP in the soil of the research area was 7.70 mg·kg$^{-1}$.

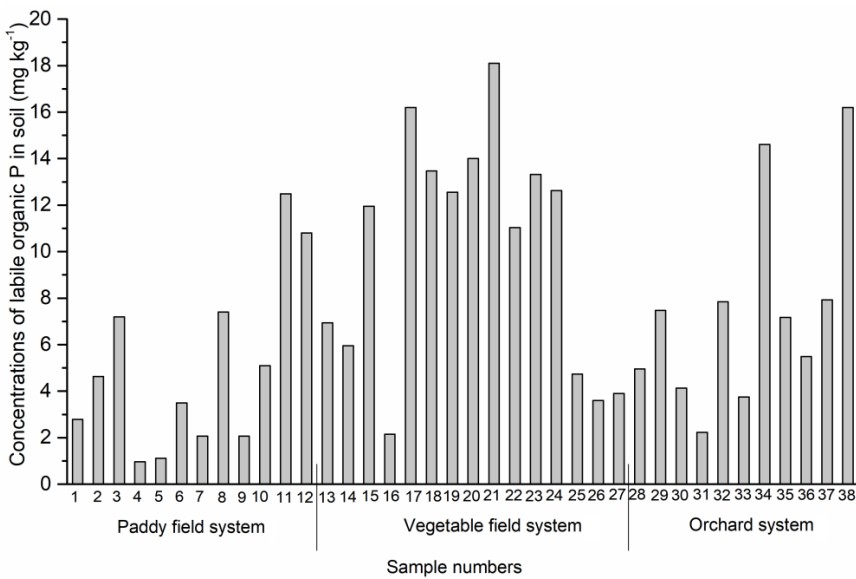

**Figure 3.** Concentrations of labile organic phosphorus in soil of different planting systems.

### 3.3.2. Moderately Labile Organic P (MLOP)

In the paddy fields, the soil MLOP concentrations varied widely, with minimum and maximum values of 12.57 and 179.31 mg·kg$^{-1}$, respectively, and a coefficient of variation of 74.3%. The average concentration of MLOP was 71.68 mg.kg$^{-1}$ in the soil of the paddy fields (Figure 4). The average concentration of the soil MLOP was 138.47 mg·kg$^{-1}$ in the vegetable fields, which is higher than that of the paddy fields. The range of vegetable fields MLOP was 32.58–258.20 mg·kg$^{-1}$, with a coefficient of variation of 44.49% (Figure 4). The orchard average MLOP concentration was 108.14 mg.kg$^{-1}$, with a range of 55.77–205.54 mg·kg$^{-1}$, and a coefficient of variation of 38.54%. The MLOP concentration in the whole research area was 108.60 mg·kg$^{-1}$ (Figure 4).

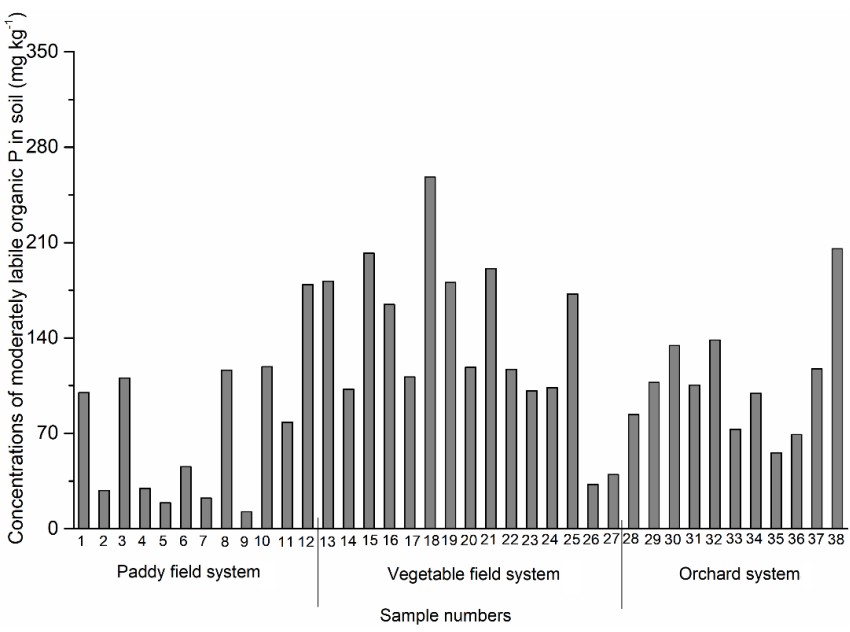

**Figure 4.** Concentrations of moderately labile organic phosphorus in soil of different planting systems.

### 3.3.3. Moderately Resistant Organic P (MROP)

The average concentration of MROP was 56.2, 29.1 and 9.74 mg·kg$^{-1}$, with variation ranges of 5.66–97.37, 2.59–69.71, and 70.0–20.6 mg·kg$^{-1}$ in the paddy fields, vegetable fields

and orchards, respectively (Figure 5). The MROP trend showed paddy fields > vegetable fields > orchards. The concentrations of MROP in the paddy fields, vegetable fields and orchards displayed similar coefficients of variation, of 68.84, 68.0 and 66.52%, respectively. The soil MROP of the area had an average of 32.05 mg.kg$^{-1}$ (Figure 5).

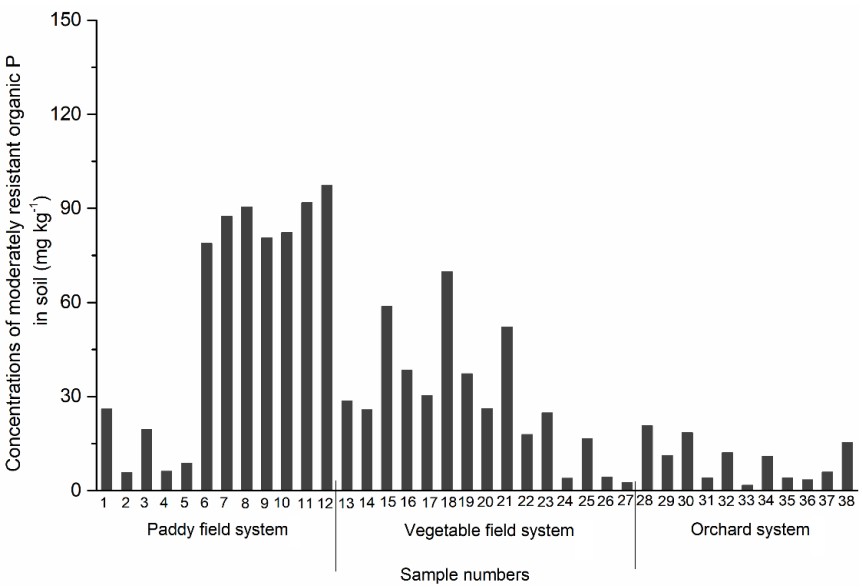

**Figure 5.** Concentrations of moderately resistant organic phosphorus in soil of different planting systems.

### 3.3.4. Highly Resistant Organic P (HROP)

The HROP concentration in the soil of three planting systems were displayed in Figure 6. The highest HROP concentration (average value 36.83 mg·kg$^{-1}$) was found in the paddy field, followed by the vegetable field (22.48 mg·kg$^{-1}$), and then the orchard system (11.99 mg·kg$^{-1}$). HROP concentration varied between different planting systems, i.e., 9.51 to 56.9 mg·kg$^{-1}$ with coefficient of variation of 35.76% in the paddy field, 3.1 to 43.97 mg·kg$^{-1}$ with coefficient of variation of 57.13% in the vegetable field, and 1.57 to 25.25 mg·kg$^{-1}$ with coefficient of variation of 61.64%, in the orchard system. In the research area, soil HROP mean concentration was 14.81 mg·kg$^{-1}$(Figure 6).

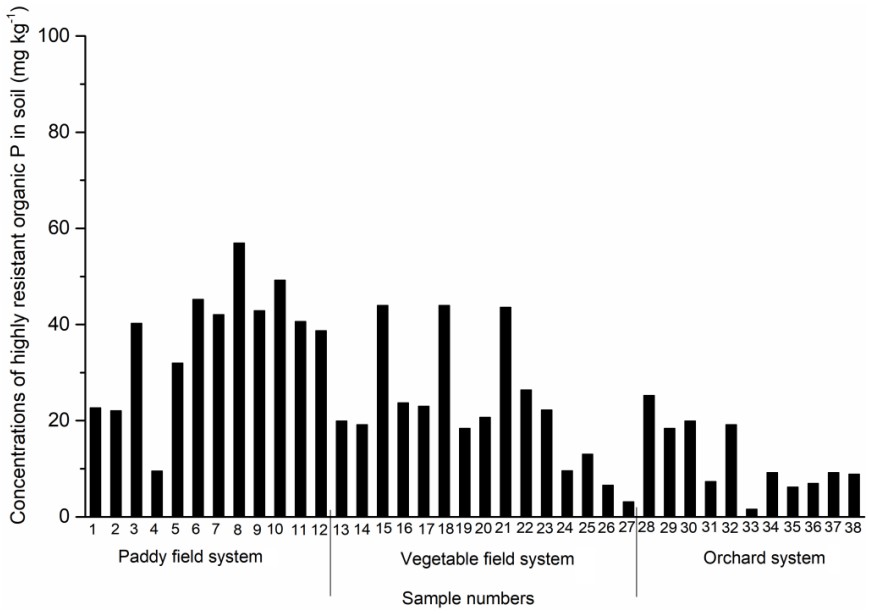

**Figure 6.** Concentrations of high stable organic phosphorus in soil of different planting systems.

*3.4. Proportion of Organic P Fractions in P Pool*

3.4.1. Proportion of Total Organic P to Total P

In different planting systems, the proportion of soil TOP in TP was lower than 50% (Figure 7). In the paddy field, soil TOP accounted for 17.2 to 47.8% of TP, with an average of 28.8%. In the vegetable field, the percentage of 16.0% was lower than that of the paddy field, ranging from 7.8 to 24.8%. The proportion of TOP to TP varied from 10.5 to 41.7%, with an average of 21.4% in the orchard, which was between that of the paddy field and the vegetable field. In the surveyed area, the proportion of TOP to TP was 21.4% (Figure 7).

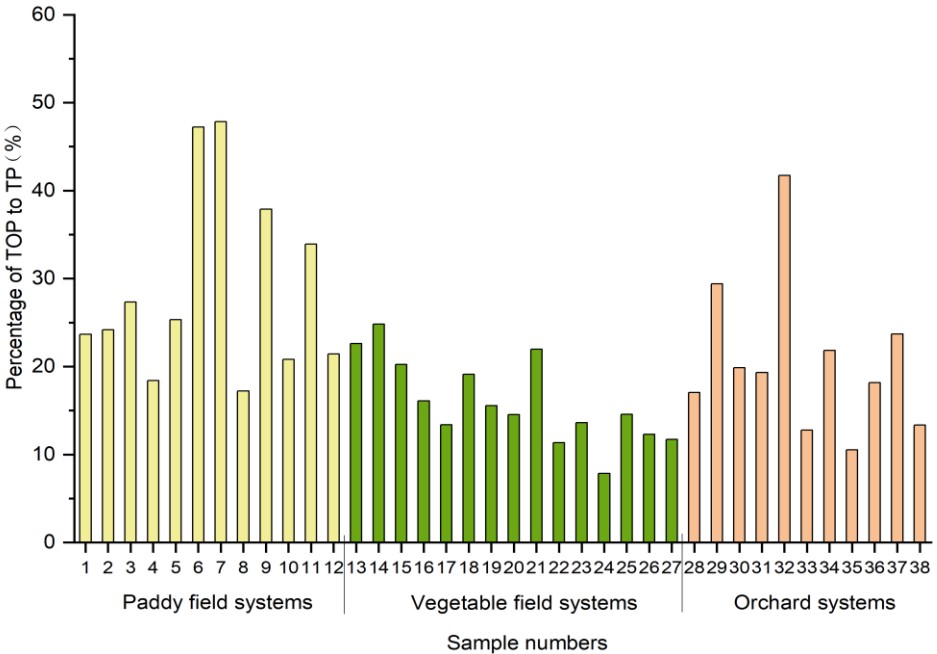

**Figure 7.** Percentage of organic phosphorus to total phosphorus in soil of different planting systems. Note: TP, total phosphorus; TOP, total organic phosphorus.

3.4.2. Proportion of Organic P Fractions

In the paddy field, MLOP dominated the total organic P pool; an average percentage of 41.6% was recorded (Figure 8A), and the percentage ranged from 9.1 to 66%, as displayed in Figure 9. A percentage of 30.2% with a variation range of 9.4–58.3% in MROP to TOP, and 25.19% with a variation range of 11.9–52.6% in HROP to TOP were observed in this planting system (Figures 8A and 9), respectively. The proportion of LOP to TOP varied from 1.4 to 7.7% in the paddy field (Figure 9), with a mean proportion of 3% obtained. The soil organic P fractions in TOP was in the following order: MLOP > MROP > HROP > LOP.

Similar to that of the paddy fields, in the vegetable fields, the main fraction in the TOP was MLOP. The range of MLOP to TOP was 61.6–83.4% (Figure 9), with an average of 70.2% (Figure 8B). Additionally, a percentage of 13.1% in MROP toTOP was observed, and 11.0% in HROP to TOP. The LOP accounted for the lowest proportion in the TOP, with an average percentage of 5.7%, and a variation range of 0.9–9.8% (Figures 8B and 9).

In the orchard soil, most of the TOP was MLOP, with a mean proportion of MLOP to TOP of 79.1%, and a range of 62.2–91.2% (Figures 8C and 9). There was a similar proportion of MROP to TOP (6.7%) and HROP to TOP (8.7%), with similar variation ranges of 2.1–15.3% and 1.96–18.8%, respectively. The LOP accounted for 5.6% in the TOP, with a percentage range of 1.87–10.90% (Figures 8C and 9).

Across the different planting systems, it was indicated that MLOP dominated the TOP pool (63.7%), followed by MROP (16.6%), HROP (14.8%), and that LOP accounted for a minor part in the TOP (4.8%).

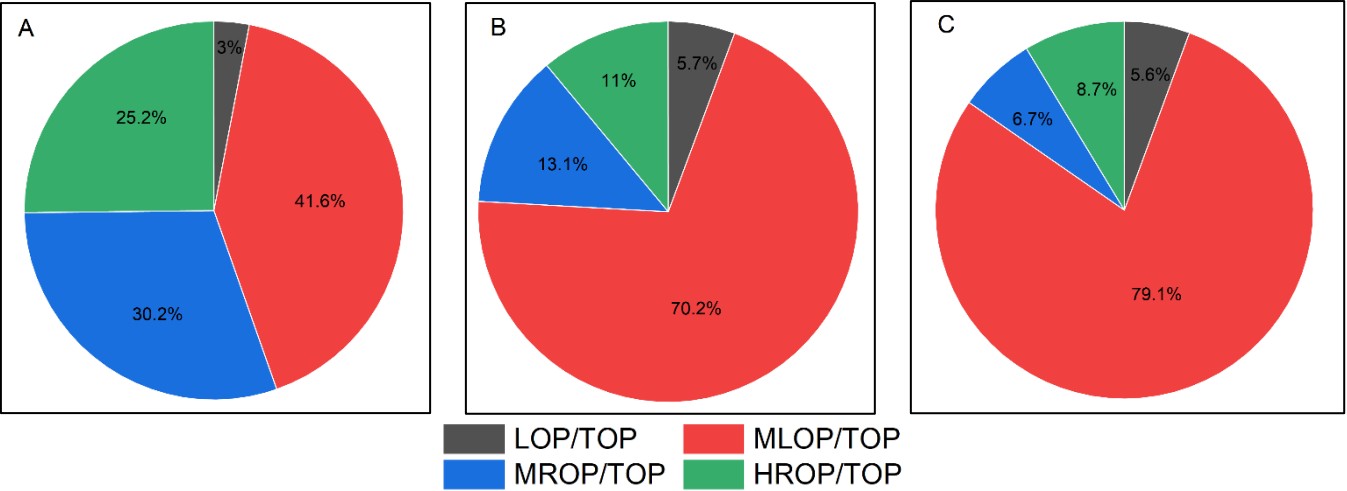

**Figure 8.** Percentage of different organic phosphorus fractions in total organic phosphorus in soil of different planting systems. ((**A**), paddy field system; (**B**), vegetable field system; (**C**), orchard system). Note: LOP, labile organic phosphorus; MLOP, moderately labile organic phosphorus; MROP, moderately resistant organic phosphorus; HROP, highly resistant organic phosphorus; TOP, total organic phosphorus.

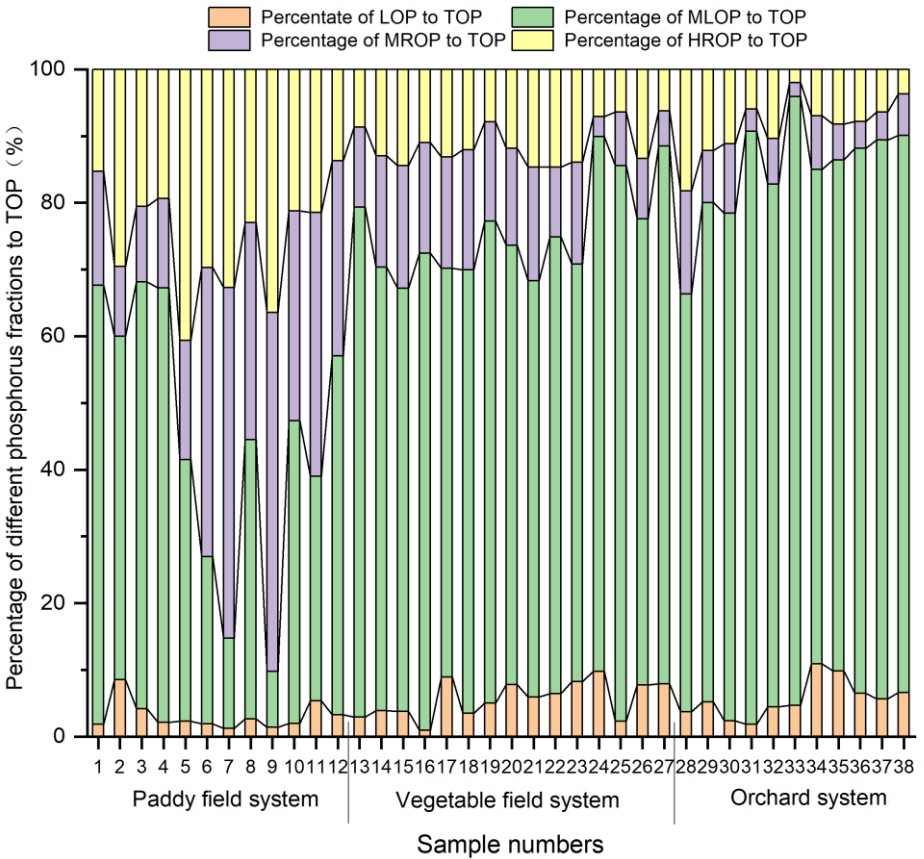

**Figure 9.** Percentage of different organic phosphorus fractions in total organic phosphorus in soil of different planting systems. Note: LOP, labile organic phosphorus; MLOP, moderately labile organic phosphorus; MROP, moderately resistant organic phosphorus; HROP, highly resistant organic phosphorus; TOP, total organic phosphorus.

### 3.5. Correlations between Soil Organic P Concentrations and Soil Properties

Correlations between the soil chemical properties and organic P fractions are displayed in Figure 10. A Pearson's correlation analysis revealed that the soil pH correlated negatively with TP, AP, LOP and MLOP. However, the soil OM correlated negatively with TP, AP, LOP and MLOP. Similarly, positive correlations also existed between soil available N and TP, AP, LOP and MLOP. Additionally, significant correlations were also detected between soil AK, TP, and AP. However, the soil AK showed no significant correlation with organic P (Figure 10).

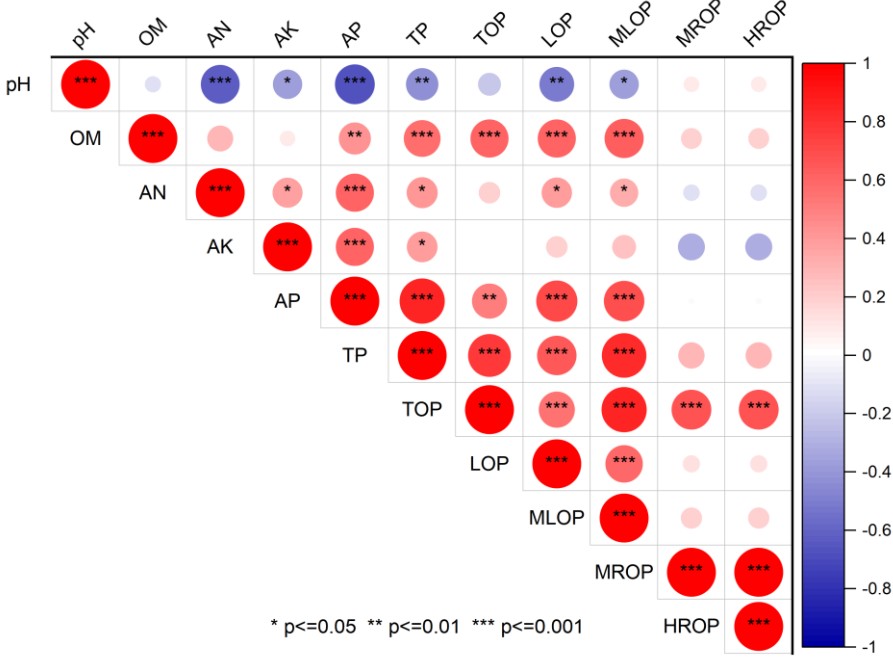

**Figure 10.** Correlation matrix between soil properties (*n* = 38) *. Note: OM, organic matter; AN, available N; AK, available K; AP, available phosphorus; TP, total phosphorus; TOP, total organic phosphorus; LOP, labile organic phosphorus; MLOP, moderately labile organic phosphorus; MROP, moderately resistant organic phosphorus; HROP, highly resistant organic phosphorus.

## 4. Discussion

The research area in this study is in the suburbs of Guangzhou city, within the Guangdong Province of South China, which is situated in the latosolic red soil zone [18]. Latosolic red soil (classified as a Ultisol [19]) was developed primarily from the underlying parent materials of granite and sandy shale. It is characterized by a low pH (~5.0), low base cations (0.3–4.91 meq·100 g$^{-1}$ soil), and low nutrient levels [18,20]. The acid soil (i.e., pH < 6.5) in different planting systems of this study was detected, indicating the chemical property of latosolic red soil. However, some farmers in the research area usually apply lime to improve the soil conditions, leading to an increase in soil pH, which was observed in some sampling sites. According to the data of the Second National Soil Census in China (in 1980) [18], concentrations of 2.35 ± 0.301% (equivalent to 23.5 ± 3.01 g·kg$^{-1}$) in OM, 105.5 ± 11.44 mg·kg$^{-1}$ in AN and 74.2 ± 1.24 mg.kg$^{-1}$ in AK in the soil of the farmland of Guangdong Province, were recorded. It was indicated that concentrations of soil available N and K in the research area increased by 22% and 105%, respectively, over the last four decades, while soil OM level showed no significant change. The results suggested that soil mineral element contents were improved under long-term local cultivation practices. This phenomenon can also be verified by the considerable accumulations of TP and AP in the soil of this study. The accumulation of P in soil may be caused by two aspects, which are the intrinsic soil qualities and anthropic activities. On the one hand, latosolic red soil has a strong phosphate-fixing capacity because of its high concentrations of Fe$_2$O$_3$

(5.06–17.49%) and $Al_2O_3$ (28.17–37.76%) [18,21], contributing to the build-up of the soil P pool. On the other hand, excessive chemical fertilizers were usually applied to rice, vegetables and the fruit trees to gain higher yields and profits, and inevitably resulted in the accumulations of P in soil [22–24]. According to the statistical data for Guangdong Province [25], phosphate fertilizer applied to farmland reached 485,400 tons in 2020, which is 3.2 times of that of 1980. Specifically, vegetable fields received the highest annual nutrient $(N + P_2O_5 + K_2O)$ input (1639.5 $kg \cdot hm^{-2}$), followed by orchards (1299 $kg \cdot hm^{-2}$), and then paddy fields (906 $kg \cdot hm^{-2}$) [26]. This nutrient management model led to discrepancies in soil P accumulation between the paddy fields, vegetable fields, and the orchards.

In most of the farmland soils, the proportion of organic P in the total P pool are lower than 50% [27]. According to Maranguit et al. (2017) [28], organic P in a tropical weathered soil was 15–40% of the total P. In black soil, albic soil, chernozem soil, dark brown soil, and loessal soil [29,30], the percentage of organic P in the total P had a range of 20–50%. Similarly, we also found that organic P accounted for a minor percentage in the soil total P pool in the current study. However, the composition of organic P fractions was considered as the key factor for evaluating the P bioavailability in the soil. LOP is considered as biologically available P, and readily mineralizable to contribute to plant P nutrition [11,31], MLOP is easily mineralized organic P forms [32], MROP (fulvic-acid Po) is not prone to be mineralized by microbes or absorbed by plants, and HROP is associated with stable organic constituents, and can be considered as a long-term P resource. Concentrations of organic P fractions in the soil of three planting systems were also found to be analogous to those in the previous works, i.e., LOP in Mollisol soil (2.93–3.8 $mg \cdot kg^{-1}$), vertisol soil (1.18–2.35 $mg \cdot kg^{-1}$) [33] and desert aeolian sandy soil (11.5–20.6 $mg \cdot kg^{-1}$) [34], MLOP in Planosol soil (10.99–43.06 $mg \cdot kg^{-1}$) [35] and loessal soil (32.83–41.95 $mg \cdot kg^{-1}$) [30], MROP in noncalcareous soils and five calcareous soils (17–52 $mg \cdot kg^{-1}$) [36], and HROP in yellow soil (11.07–46.96 $mg \cdot kg^{-1}$) [37], silt loam calcareous soil (1.83–6.58 $mg \cdot kg^{-1}$) [38], and paddy soil (30.9–35.1 $mg \cdot kg^{-1}$) [39]. In this study, the MLOP dominated the organic phosphorus pool, irrespective of planting systems, indicating that organic P is an important potential source of bioavailable P in the research area. The LOP had the lowest proportion in the total organic P pool, which may reflect P adsorption and immobilization, owing to the dominance of goethite, kaolinite, and hematite in these soils [18,40]. The distribution of organic P fractions in the soil of this study was in agreement with the findings of He et al. [41], Danilo and Ibanor [42], and Sharpey and Smith [36]. However, it was inconsistent with the results of Li et al. [43], where HROP was dominant, with the highest content in the organic P pool. It has been reported that long-term fertilization can significantly change the content and composition of P fractions in the soil [44,45]. Specifically, the long-term addition of P made a larger contribution to labile and moderately labile P, than to non-labile P [2]. The different strategies of nutrient applications in rice, vegetable and fruit cultivation may contribute to the variations in the soil organic phosphorus pools in the three planting systems [26].

As mentioned above, soil organic P accumulation is usually affected by soil properties. The negative correlations between soil pH and organic P (LOP and MLOP) may indicate that soil pH can regulate the interaction between organic P compounds and clay minerals, by affecting the cation exchange sites [46,47]. Furthermore, a low soil pH may provide optimum conditions for enzyme activity and microbe reproduction, facilitating accumulations of some organic P fractions (such as Inositol hexakisphosphate) [48,49]. In contrast, soil OM showed a positive correlation with LOP, MLOP, TP and AP, which was also reported in previous studies [50,51]. This is because carbon is the main energy provider during organophosphorus synthesis from soil microbial sources [52], and in addition, organic matter in the soil probably induces a decrease in P adsorption [53]. Both help to illustrate the interactions between soil organic carbon and the P pool. Additionally, interactions between N and P cycling in soil have been extensively confirmed [54–56]. Nitrogen influences P cycling via nitrogen-bearing phosphatases, the soil pH during the nitrification process, and the N/P stoichiometry of microorganisms [57–59]. The positive correlations between

available N and organic P in the soil of this study agree with the findings of Wang et al. (2021) [47] and Margalef et al. (2017) [60].

## 5. Conclusions

The results obtained from the current study indicate that over the past four decades, an excessive amount of P has accumulated in the planting soil in the suburban area of Guangzhou city, Guangdong Province, South China. The soil organic P concentration varied widely among different planting systems, with a relatively higher level observed in the vegetable fields, followed by the paddy fields and then the orchards. Soil MLOP was the dominant class of organic P in these soils, indicating that organic P may be an important potential source of bio-available P. The soil chemical properties, including pH, organic matter and available N, are key factors affecting the organic P pool. From the perspective of precision management of the soil P nutrients, further studies are warranted, to illustrate the components of labile organic phosphorus and moderately labile organic phosphorus, by using a non-wet chemical analysis method e.g., solution $^{31}$P-nuclear magnetic resonance (NMR) spectroscopy, and then evaluating the bio-availability of these organic compounds.

**Supplementary Materials:** The following supporting information can be downloaded at: https://www.mdpi.com/article/10.3390/horticulturae8111055/s1, Table S1: Classification of soil nutrient concentrations.

**Author Contributions:** Conceptualization, T.L. and J.N.; methodology, J.N. and Y.C.; software, L.H.; formal analysis, T.L. and Y.C.; resources, J.Y. and J.N.; data curation, R.Z.; writing—original draft preparation, T.L.; Review and editing, M.Z.; writing—review and editing, J.N.; All authors have read and agreed to the published version of the manuscript.

**Funding:** This research was jointly funded by the National Natural Science Foundation of China (31701996), the Science and Technology Planning Project of Guangdong Province, China (2021B1212050019) and low carbon agriculture and carbon neutralization Research Center, GDAAS (XTXM202204).

**Data Availability Statement:** Tong Li and Jiangfeng Ning are responsible for data keeping, and data are available upon request.

**Conflicts of Interest:** The authors declare no conflict of interest.

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
