# Peer review of "Characteristics of Organic Phosphorus Pool in Soil of Typical Agriculture Systems in South China"

_horticulturae, doi:10.3390/horticulturae8111055_

Round 1

Reviewer 1 Report

The presented study is of great practical importance because it shows the content of one of the main macronutrients in the soil in southern China. The analysis of phosphorus content on the basis of 38 samples in three different crane systems allowed for a reliable assessment of the level of this component. Detailed measurements and the obtained values allow for optimal and rational fertilization with these components. There are few editorial errors in the work, e.g.

- Table 1 should present the units of measurement with the applicable standards, including the SI system.

- In Fig.1, in line 200, the captions in the figure should be completed, as they are incomplete in their present form.

Author Response

Response to Reviewer 1:  

Thanks for your comments on our paper. We have revised the paper according to your comments:  

Question 1. 

Table 1 should present the units of measurement with the applicable standards, including the SI system.

Response:

It is our mistakes that the data in the Table 1 with the required template is not fully displayed. Table 1 has been edited and the data information including the parameter description and the unit can be showed clearly.   

Question 2.

In Fig.1, in line 200, the captions in the figure should be completed, as they are incomplete in their present form.

Response:

The captions of figures in this manuscript have all been checked and improved.

Reviewer 2 Report

Dear Authors,

I am glad that I was selected as a reviewer of this interesting manuscript entitled: "Characteristics of organic phosphorus pool in soil of typical agriculture systems in South China" by Tong Li, Jianwu Yao, Ruikun Zeng, Yong Chen, Lijiang Hu and Jianfeng Ning.

The purpose of the research presented in this manuscript was to investigate the current status of soil total phosphorus and available phosphorus in paddy fields, vegetable fields and orchards. In addition, the Authors also decided to clarify the soil organic phosphorus pool, and its distribution in different agricultural environments.

The results of this research will allow for the sustainable management of this essential macroelement for plants, and thus the protection of phosphorus resources. Therefore, the undertaken research topic seems to be very important, especially in the light of the intensive development of sustainable agriculture.

The research methodology is well presented in this manuscript. The discussion is very interesting. I really like the Conclusions section.

However, I have a few comments that the Authors should consider at a later stage in the preparation of the manuscript for printing:

1) There is no hypothesis in the Abstract, please complete.

2) Introduction is based on only 10 publications. It seems to me that the Authors should cite more publications here.

3) There is a lot of ambiguity in the Results section, mainly due to technical problems:

a) Table 1 - what does it mean "Pa-", "Organic mat-", "Available" and "C.V. (";

b) Figure 2 - under the figure, explain the meaning of abbreviations: TP, AP and TOP; please remember that tables and figures should be understandable without looking for an explanation of the meaning of abbreviations in the text;

c) Figure 8 - under the figure, explain the meaning of the abbreviations: LOP / TOP, MLOP / TOP, MROP / TOP and HROP / TOP; moreover, the colors for LOP / TOP, MLOP / TOP, MROP / TOP and HROP / TOP on the charts for individual "fiels systems" should be the same;

d) Figure 9 - under the figure, explain the meaning of abbreviations: LOP, TOP, MROP, MLOP and HROP.

5) The entire manuscript should be carefully adapted to the template in the Horticulturae journal.

Overall, I believe that, after considering my comments, the research presented in this manuscript merits publication in the journal Horticulturae.

Author Response

Response to Reviewer 2:  

Thanks for your comments and we have revised the paper according to your comments:  

Question 1.

There is no hypothesis in the Abstract, please complete.

Response:

The hypothesis of this study has been added in the abstract:  

It was hypothesized that soil organic P pool distinguished each other between different land utilization pattern.

Question 2.

Introduction is based on only 10 publications. It seems to me that the Authors should cite more publications here.

Response: In order to elucidate the research background of this study, four relevant references were added in the Introduction section in this manuscript. Meanwhile, the newly added literatures in whole text and the reference list have been improved.

Question 3.

3) There is a lot of ambiguity in the Results section, mainly due to technical problems:

  1. a) Table 1 - what does it mean "Pa-", "Organic mat-", "Available" and "C.V. (";
  2. b) Figure 2 - under the figure, explain the meaning of abbreviations: TP, AP and TOP; please remember that tables and figures should be understandable without looking for an explanation of the meaning of abbreviations in the text;
  3. c) Figure 8 - under the figure, explain the meaning of the abbreviations: LOP / TOP, MLOP / TOP, MROP / TOP and HROP / TOP; moreover, the colors for LOP / TOP, MLOP / TOP, MROP / TOP and HROP / TOP on the charts for individual "fiels systems" should be the same;
  4. d) Figure 9 - under the figure, explain the meaning of abbreviations: LOP, TOP, MROP, MLOP and HROP.

Response:

  1. The information of Table 1 has been improved.
  2. The explanation of abbreviations under all figures has been checked and added with necessity in the whole text.
  3. The figure 8 has been revised according to the reviewer’s suggestions.
  4. The explanation of abbreviations: LOP, TOP, MROP, MLOP and HROP under figure 9 has been added.

Question 4.

  • The entire manuscript should be carefully adapted to the template in the Horticulturae journal.

Response: The whole text has been edited according to the template of Horticulturea.

Reviewer 3 Report

The study would be complete in this research if the author had non-agricultural soils, such as forest soils nearby the study site, as comparison soil.

Author Response

Response to Reviewer 3:  

Thanks for your comments and we have revised the paper according to your comments:  

Question 1.

The study would be complete in this research if the author had non-agricultural soils, such as forest soils nearby the study site, as comparison soil.

Response: It is a very good suggestion. In this manuscript, we focused on the soil of agricultural land. In the future, if it were feasible, we will try our best to make a comprehensive research focusing on various terrestrial ecosystem, i.e.farmlands, forests as well as the grasslands.

Questions in the manuscript.

  • Line 12: The word of “in”in the abstract.

Response: It has been revised as “of”.

  • Line 106: Why is the number of representative samples in each area not the same?

Please give more information about the criteria/reason for site selection in each study area.

Response: The background related to the soil site selected in this study has been added:  

A part of farmland in the research area is used to be cultivated with rotation model, i.e. rice and vegetables, potato or other crops rotation annually. In order to explore the influence of typical agriculture practices on soil organic pool, the farmland with following features were selected in this study: annual tillage with single planting mode for more than 5 years, i.e. paddy field with double cropping rice (early rice and late rice), vegetables field with year-round vegetable cultivation, and the orchard planted with perennial fruit trees.

  • Line 119: potassium dichromate

Response: The two phrases of “dichromate” and “bichromate” are not identical in meaning. We would like to adopt the reviewer’s comment.

  • Line 122: exchangeable potassium

Response: In soil science and plant nutrition field, the description of “available potassium” is widely accepted and used. In this manuscript, we focused on the P nutrition in soil, so we would like to use the usus loquendi.

  • Line 125: HCIO4

Response: It has been revised as a correct format of HCIO4.

  • Line 164: Table 1 data information

Response: It has been improved.

  • Line 315:mEq

Response: It has been revised as meq.

  • Line 320: please change the unit of OM to the same team as shown in table 1

Response: It has been improved and revised.

  • Line 363:highest

Response: It has been revised as the highest.

  • Line 396, 397:Po

Response: It has been revised as the organic phosphorus.